# Diagnosing Model Editing via Knowledge Spectrum

## Abstract

Model editing, the process of efficiently modifying factual knowledge in pre-trained language models, is critical for maintaining their accuracy and relevance. However, existing editing methods often introduce unintended side effects, degrading model performance in unpredictable ways. While much research has focused on improving editing algorithms, the role of the target knowledge's intrinsic properties remains a significant, underexplored factor. This paper addresses this gap by first proposing the "Knowledge Spectrum," a systematic framework for categorizing knowledge based on its real-world popularity, the model's pre-edit familiarity, and the linguistic structure of the eliciting question. Our empirical analysis reveals that these characteristics are strong predictors of editing success and stability. Informed by these findings, we introduce the "Knowledge-Diagnostic Framework," an adaptive strategy that tailors editing intensity to the diagnosed difficulty of a knowledge item. We demonstrate that this framework significantly improves success rates for challenging edits while optimizing computational resources. Our work provides a more comprehensive understanding of the factors governing model editing.

## 1 Introduction

Large language models (LLMs) are increasingly deployed as knowledge-intensive systems, yet their parametric knowledge inevitably lags behind a changing world. Model editing—updating a model's internal parameters to correct or insert facts without full retraining—has therefore become a practical necessity Yao et al. (2023). Despite rapid progress, edits can introduce unintended side effects, harming unrelated knowledge or degrading general reasoning Gu et al. (2024); Yang et al. (2024). Most prior work addresses these risks by improving how we *apply* an edit. In contrast, comparatively little is known about how the *nature of the target knowledge itself* shapes the difficulty and safety of editing.

This paper closes that gap. We ask: when does an edit tend to succeed cleanly, and when is it inherently brittle? Common intuition suggests that not all facts are equal: some are prominent in training data, some are already correctly (or incorrectly) entrenched in the model, and prompts differ in the reasoning they elicit. We make this intuition operational by introducing the **Knowledge Spectrum**, a simple three-dimensional lens for categorizing target knowledge along: (i) **Popularity** (a proxy for exposure in pretraining, measured by real-world signals such as Wikipedia page views); (ii) **Familiarity** (whether the model already knows the fact pre-edit, inspired by SliCK-style probing Gekhman et al. (2024)); and (iii) **Question Type** (the syntactic form of the eliciting prompt, e.g., *why* vs. *which*). This framing lets us move beyond one-size-fits-all editing and quantify which kinds of targets are intrinsically harder or riskier.

Our analysis surfaces three robust regularities. First, inserting *unknown* facts is consistently easier and safer than overwriting *known* ones, indicating resistance from entrenched representations. Second, edits involving *famous* entities (high popularity) succeed more often than those about obscure entities, consistent with clearer, more localizable internal memories. Third, *question type matters*: *which*-style prompts are the most brittle, while *why*-style prompts are comparatively forgiving, suggesting different editing pressure on discrete vs. explanatory representations. Notably, AlphaEdit's null-space projection provides strong stability across these conditions, but difficulty patterns persist.

Guided by these findings, we propose the **Knowledge-Diagnostic Editing Framework**. A lightweight *diagnostic engine* first classifies a target along the Knowledge Spectrum. The editor then adapts its intensity: difficult targets (e.g., known, unfamous, or *which*-type) receive a stronger intervention (e.g., repeated AlphaEdit passes), while easy targets receive a single pass. This simple policy improves success on hard cases and saves compute on easy ones, yielding substantial end-to-end efficiency gains without sacrificing stability.

In sum, our contribution is threefold. First, we advance a knowledge-centric view of model editing by introducing the *Knowledge Spectrum*—capturing popularity, familiarity, and question type—and showing that these axes reliably predict both editing efficacy and side effects. Second, we broaden evaluation beyond locality by combining reliability and generalization with general-ability benchmarks, uncovering degradations invisible to locality-only tests. Finally, we propose the *Knowledge-Diagnostic Editing Framework*, an adaptive approach that adjusts edit intensity according to diagnosed difficulty, thereby improving success on hard cases while saving compute on easy ones. Taken together, our results reframe model editing as a *knowledge-aware* process: the right algorithm matters, but so does the kind of knowledge being changed. Designing editors and evaluations that account for this structure is key to making edits both reliable and economical.

## 2 RELATED WORK

Model editing aims to efficiently update the knowledge within a pre-trained language model without the substantial cost of full retraining. The field has rapidly evolved, yielding a variety of techniques that can be broadly classified into two main paradigms based on their interaction with the model's original parameters Yao et al. (2023); Wang et al. (2023).

The first paradigm, **parameter-preserving methods**, avoids altering the weights of the base LLM. Instead, new knowledge is introduced by augmenting the model with external components or new, isolated parameters. Memory-augmented approaches, for example, store updated facts in an external datastore. When presented with a relevant query, a retriever fetches the correct information to guide the model's generation. A prominent example is SERAC, which employs a separate, smaller "patch" model to handle edited facts and a classifier to determine whether to invoke the original or the patch model for a given input Mitchell et al. (2022). Another strategy involves freezing the original model's weights and inserting a small number of new, trainable parameters, often in the form of "adapter" layers, which are specifically trained to encapsulate the new knowledge Hartvigsen et al. (2024); Yu et al. (2024). While these methods are non-invasive, they can introduce inference latency and may struggle with deep knowledge integration, as the model's core parametric knowledge remains separate from the external updates, potentially leading to knowledge conflicts Xu et al. (2024).

The second paradigm, **parameter-modifying methods**, directly intervenes in the model's internal weights to insert, alter, or erase information. This paper focuses on this category due to its potential for creating deeper, more permanent, and more efficient knowledge updates. Traditional fine-tuning is the classic approach, but it often suffers from catastrophic forgetting. To mitigate this, constrained fine-tuning methods update only a small subset of the model's parameters Zhu et al. (2021); Rafailov et al. (2023). A more precise and influential sub-category is the **locate-and-edit** paradigm. This approach is founded on the key insight from interpretability research that factual knowledge in transformers is not arbitrarily distributed but is often localized within specific feed-forward (FFN) layers, which can be conceptualized as key-value memories Geva et al. (2021). These methods first use causal analysis to locate the critical model components responsible for storing a specific fact and then perform a surgical update to modify the stored association Meng et al. (2022a). By directly altering the model's internal world representation, this approach avoids the inference latency of external modules and aims for a more profound and generalizable form of learning.

## 3 PRELIMINARY

### 3.1 CORE EDITING ALGORITHMS

Two state-of-the-art locate-and-edit algorithms are central to our investigation: MEMIT and AlphaEdit.

**MEMIT (Mass-Editing Memory in a Transformer)** is a powerful and scalable implementation of the locate-and-edit approach, capable of applying thousands of edits in a single batch process Meng et al. (2022b). Instead of concentrating an edit on a single layer, MEMIT distributes the update across several MLP layers identified during the location phase. The update is formulated as a constrained optimization problem. For a set of new key-value pairs $(K_1, V_1)$ to be inserted and a set of existing pairs $(K_0, V_0)$ to be preserved, MEMIT solves for a minimal parameter update $\Delta W$ that minimizes a joint objective, conceptually represented as:

$$\underset{\Delta}{\arg\min} \left( \|(W + \Delta)K_1 - V_1\|^2 + \lambda\|(W + \Delta)K_0 - V_0\|^2 \right)$$

The first term enforces the new knowledge, while the second acts as a regularization term to preserve existing knowledge. By providing an efficient closed-form solution to this problem, MEMIT offers a practical tool for large-scale editing.

**AlphaEdit** was developed to address the limitation that methods like MEMIT can still introduce unintended side effects, especially when edits interfere with one another. It offers a stronger mathematical guarantee of safety by employing a two-step process based on null-space projection Fang et al. (2025). First, it identifies the null-space of the preserved knowledge keys ($K_0$), a "safe" subspace where any modification is mathematically guaranteed to have zero impact on the preserved facts. It computes a projection matrix $P$ such that $PK_0 = 0$. Second, it calculates the required update for the new fact and projects it into this safe null-space before applying it. The objective is to find an update $\Delta$ that minimizes the error for the new fact, constrained entirely to this safe zone:

$$\underset{\Delta}{\arg\min} \|(W + \Delta P)K_1 - V_1\|^2$$

This approach ensures that the edit is surgically precise with respect to the specified preserved knowledge, making it particularly robust against certain forms of side effects.

### 3.2 EVALUATING MODEL EDITING: BEYOND LOCALITY

A rigorous evaluation of a model edit requires assessing its impact from multiple perspectives Yao et al. (2023). The primary success of an edit is typically measured by two core metrics: **Reliability**, which checks if the model produces the target answer for the exact edit prompt, and **Generalization**, which tests if the model can apply the new knowledge to paraphrased versions of the prompt.

However, the greatest challenge lies in evaluating unintended side effects. The mainstream approach to this has been to measure **Locality** Meng et al. (2022a). This involves testing whether an edit on a specific fact (e.g., "The Eiffel Tower is in Paris") has unintentionally altered unrelated but semantically nearby knowledge (e.g., "The Colosseum is in Rome"). While crucial, this approach has a narrow scope and may not capture more subtle or widespread forms of model degradation. Recent studies have begun to highlight that edits can harm a model's fundamental reasoning and comprehension skills, even if they pass locality tests Gu et al. (2024); Cohen et al. (2023); Yang et al. (2024).

Our work argues for and contributes to an expanded evaluation paradigm. We posit that a comprehensive assessment must also measure the impact on a model's **General Ability**. An edit might not affect other specific facts but could still damage the underlying cognitive machinery of the model. Therefore, our methodology extends beyond locality checks by testing the post-edit model's performance on a suite of standardized academic benchmarks, such as ARC Clark et al. (2018) and Open-BookQA Mihaylov et al. (2018), which are designed to probe core reasoning capabilities rather than simple fact recall. A drop in performance on these benchmarks signals a deeper and more concerning side effect.

## 4 METHODOLOGY

To systematically investigate how the intrinsic properties of knowledge affect model editing, we designed a comprehensive methodology centered around three key components. First, we formally define the editing task and its success criteria. Second, we introduce the "Knowledge Spectrum," a novel framework for classifying target knowledge. Finally, we propose the Knowledge-Diagnostic Framework, an adaptive strategy that leverages this classification to optimize editing outcomes.

## 4.1 Task Definition and Desiderata

The fundamental goal of model editing is to alter the factual knowledge within a pre-trained language model $f_\theta$ to produce an edited model $f_{\theta'}$. Formally, given an edit request represented by an input prompt $x_e$ and a desired new output $y_e$, an editing algorithm $A$ generates a set of modified parameters $\theta' = A(\theta, x_e, y_e)$. The resulting edited model, $f_{\theta'}$, must satisfy three critical desiderata to be considered successful:

- **Efficacy:** The model must successfully learn the new information. This is measured through two metrics: (1) *Reliability*, where the model must produce the target answer for the exact edit prompt ($f_{\theta'}(x_e) \rightarrow y_e$), and (2) *Generalization*, where the model must provide the same correct answer to paraphrased versions of the original prompt ($x_e'$), demonstrating a deeper understanding rather than surface-level memorization.

- **Locality:** The edit should be surgically precise, leaving the model's vast repository of unrelated knowledge unharmed. The mainstream method for measuring this is by testing a set of unrelated facts to ensure their outputs remain unchanged.

- **General Ability:** We argue that true safety extends beyond locality. An edit must not impair the model's fundamental cognitive capabilities. We measure this by evaluating the post-edit model's performance on a suite of standardized reasoning benchmarks, quantifying any degradation in its general problem-solving skills.

## 4.2 The Knowledge Spectrum: A Framework for Analysis

To move beyond a monolithic view of knowledge, we introduce the **Knowledge Spectrum**, a three-dimensional framework for characterizing any target edit. This framework allows us to dissect the challenges of editing by analyzing knowledge based on its real-world prominence, its status within the model's internal belief system, and its linguistic structure.

**Popularity** measures how well-known an entity or fact is, serving as a proxy for its likely representational strength within the model's pre-training corpus. We hypothesize that facts about famous entities, having been encountered frequently and in diverse contexts, have more robust and well-defined neural representations, making them easier to locate and edit. Conversely, obscure facts may have sparse, diffuse representations that are more difficult to modify reliably. We operationalize this dimension by using the monthly Wikipedia page views of the subject entity in a given question. Based on the distribution of these views, we bin each knowledge item into one of two categories: **Famous** (high page views) or **Unfamous** (low page views).

**Familiarity** assesses the model's internal "intellectual state" with respect to a fact **before** any edit is performed. This dimension distinguishes between overwriting an existing belief and filling a knowledge vacuum. We hypothesize that modifying a pre-existing belief, whether correct or incorrect, presents greater resistance than inserting a fact about which the model has no prior information. Inspired by the SliCK framework Gekhman et al. (2024), we measure familiarity by probing the model's ability to generate the correct answer under various decoding strategies prior to the edit. We classify knowledge into two groups: **Known**, where the model can correctly produce the target answer, implying an established neural pathway that must be altered; and **Unknown**, where the model is unable to produce the correct answer, representing a "representational void" to be filled.

**Question Type** considers the linguistic and syntactic structure of the prompt used to elicit the knowledge. Different question forms may trigger different reasoning processes or access different knowledge representations within the model. For example, a question requiring a choice from a discrete set may target a different neural circuit than one that asks for an explanation. To analyze this, we categorize the questions in our dataset into eight distinct types based on the leading interrogative word: **Who, What, When, Where, Which, Why, How,** and **Others**. This allows us to systematically investigate how variations in the editing prompt's structure impact both efficacy and side effects.

## 4.3 The Knowledge-Diagnostic Editing Framework

Our preliminary experiments confirmed that a one-size-fits-all approach to model editing is suboptimal. The success and stability of an edit are heavily contingent on the knowledge's position within

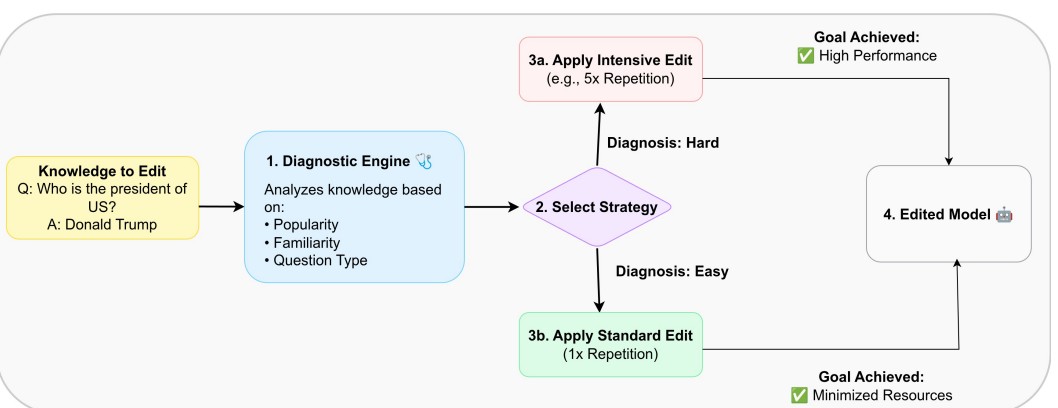

Figure 1: A conceptual illustration of our Knowledge-Diagnostic Editing Framework. It first diagnoses the target knowledge across the Knowledge Spectrum and then applies either a standard (1x) or intensive (5x) editing strategy based on the classification.

the Knowledge Spectrum. To address this, we propose the **Knowledge-Diagnostic Editing Framework**, an adaptive, two-stage strategy designed to intelligently allocate computational resources and maximize performance.

**Stage 1: The Diagnostic Engine.** When a piece of knowledge is targeted for an edit, it is first fed into our Diagnostic Engine. This engine analyzes the knowledge across the three dimensions of the Knowledge Spectrum (Popularity, Familiarity, and Question Type). Based on our empirical findings (detailed in the next section), the engine classifies the edit into one of two categories. **Hard Cases** are edits that consistently exhibit lower baseline success rates and higher risk of side effects; these include knowledge that is 'Known', 'Unfamous', or of the 'Which' question type. All other edits are classified as **Easy Cases**.

**Stage 2: Adaptive Editing Application.** Depending on the diagnosis, a tailored editing strategy is applied. For 'Hard Cases', where a standard edit is likely to fail, the framework applies an **Intensive Edit** strategy. In our experiments, this is operationalized as applying a state-of-the-art editing algorithm (AlphaEdit) multiple times (e.g., 5 repetitions). The explicit goal is to provide sufficient "force" to overcome the inherent resistance of these difficult edits. For "Easy Cases", where a single edit is likely to succeed, the framework applies a **Standard Edit** (1 repetition). This adaptive application of resources allows the framework to improve overall efficacy by focusing effort where it is most needed, while simultaneously enhancing efficiency by avoiding unnecessary computation on edits that are already likely to succeed.

## 4.4 DATASETS AND EVALUATION

The primary dataset for our general analysis is **RealTimeQA** Kasai et al. (2022), which contains a continuous stream of time-sensitive questions sourced from weekly news outlets. Unlike static benchmarks, it mirrors the real-world need to keep LLMs updated. We preprocess the original multiple-choice format into a standardized structure suitable for editing, containing fields for the question, subject, answer, and a human-written rephrased question for testing generalization.

Our evaluation protocol measures both efficacy and side effects. Efficacy is assessed via **Reliability** (accuracy on the original question) and **Generalization** (accuracy on the rephrased question). Side effects are measured by testing for degradation in **General Ability**, using the average performance across standardized reasoning benchmarks, including ARC Clark et al. (2018) and OpenBookQA Mihaylov et al. (2018), before and after editing.

| Method | Editing Success Rate | | | | General Ability | | | |
| --- | --- | --- | --- | --- | --- | --- | --- | --- |
| | LLaMA-3.1 (8B) | | LLaMA-3.2 (3B) | | LLaMA-3.1 (8B) | | LLaMA-3.2 (3B) | |
| | Known | Unknown | Known | Unknown | Known | Unknown | Known | Unknown |
| FT | 0.30 | **0.35** | 0.42 | **0.48** | 0.33 | **0.37** | 0.36 | **0.37** |
| MEMIT | 0.42 | **0.49** | 0.67 | **0.73** | 0.45 | **0.50** | 0.45 | **0.47** |
| AlphaEdit | 0.84 | **0.88** | 0.82 | **0.87** | **0.55** | **0.55** | **0.49** | **0.49** |

Table 1: Comparison of **Editing Success Rate** (left) and **Post-edit General Ability** (right) for **Known** vs. **Unknown** knowledge across LLaMA-3.1 (8B) and LLaMA-3.2 (3B).

| Method | Editing Success Rate | | | | General Ability | | | |
| --- | --- | --- | --- | --- | --- | --- | --- | --- |
| | LLaMA-3.1 (8B) | | LLaMA-3.2 (3B) | | LLaMA-3.1 (8B) | | LLaMA-3.2 (3B) | |
| | Famous | Unfamous | Famous | Unfamous | Famous | Unfamous | Famous | Unfamous |
| FT | **0.36** | 0.31 | **0.54** | 0.45 | **0.37** | 0.33 | **0.37** | 0.34 |
| MEMIT | **0.48** | 0.42 | **0.64** | 0.55 | **0.41** | 0.36 | **0.46** | 0.45 |
| AlphaEdit | **0.88** | 0.82 | **0.82** | 0.76 | **0.55** | **0.55** | **0.49** | **0.49** |

Table 2: Comparison of **Editing Success Rate** (left) and **Post-edit General Ability** (right) for **Famous** vs. **Unfamous** knowledge across LLaMA-3.1 (8B) and LLaMA-3.2 (3B).

## 5 THE IMPACT OF KNOWLEDGE CHARACTERISTICS

### 5.1 IMPACT OF FAMILIARITY: KNOWN VS. UNKNOWN.

We first investigate whether it is more challenging to modify a belief the model already holds (**Known**) versus inserting a completely new fact (**Unknown**).

The results in Table 1 show a clear and consistent trend: for every model and editing method, the success rate for editing **Unknown** knowledge is higher than for **Known** knowledge. For instance, using AlphaEdit on LLaMA-3.1, the success rate for Unknown facts is 0.88, compared to 0.84 for Known facts. This performance gap suggests that overwriting a pre-existing, and potentially entrenched, neural representation faces more resistance than establishing a new representation in a relative "void."

Table 1 further illuminates the risks. For less precise methods like FT and MEMIT, editing **Known** knowledge is demonstrably more disruptive to the model's general abilities. This implies that the process of overwriting an established belief carries a higher risk of collateral damage to adjacent or underlying reasoning structures. Notably, AlphaEdit exhibits remarkable stability; its null-space projection mechanism appears highly effective at isolating the edit, resulting in identical General Ability scores regardless of the knowledge's familiarity. This indicates a higher degree of safety in terms of preserving general capabilities, a property we will revisit later.

### 5.2 IMPACT OF POPULARITY: FAMOUS VS. UNFAMOUS.

Next, we examine whether editing facts about well-known (**Famous**) entities is different from editing those about obscure (**Unfamous**) ones. We hypothesize that an entity's prominence in the training data correlates with the clarity of its neural representation. The results are shown in Table 2.

A similarly consistent pattern emerges: editing **Famous** knowledge yields a higher success rate across all conditions. Using AlphaEdit on LLaMA-3.1, the rate for famous facts is 0.88, dropping to 0.82 for unfamous ones. This finding supports the hypothesis that the robustness of a fact's internal representation is a key determinant of its editability. Locate-and-edit algorithms can more easily pinpoint and modify the well-defined neural pathways associated with famous entities.

In terms of side effects, Table 2 shows that for FT and MEMIT, editing less-defined **Unfamous** knowledge is slightly more disruptive. This may indicate that modifications to weaker or more diffuse representations have a higher tendency to cause unintended interference. Again, AlphaEdit's score remains constant across both categories, reinforcing the conclusion that its projection mecha-

| | Editing Success Rate | | | | | | | | General Ability | | | | | | | |
|---|---|---|---|---|---|---|---|---|---|---|---|---|---|---|---|---|
| Method | why | which | who | what | when | where | how | others | why | which | who | what | when | where | how | others |
| FT | 0.41 | 0.35 | 0.41 | 0.35 | 0.35 | 0.36 | 0.41 | 0.40 | 0.46 | 0.28 | 0.36 | 0.31 | 0.37 | 0.35 | 0.41 | 0.35 |
| MEMIT | 0.64 | 0.39 | 0.64 | 0.55 | 0.46 | 0.43 | 0.64 | 0.53 | 0.55 | 0.37 | 0.41 | 0.49 | 0.54 | 0.49 | 0.39 | 0.54 |
| AlphaEdit | 0.79 | 0.70 | 0.79 | 0.75 | 0.75 | 0.73 | 0.79 | 0.75 | 0.55 | 0.55 | 0.55 | 0.55 | 0.55 | 0.55 | 0.55 | 0.55 |

Table 3: Performance of LLaMA-3.1 (8B) across different question types (reordered). Left: Editing Success Rate. Right: General Ability.

| | Editing Success Rate | | | | | | | | General Ability | | | | | | | |
|---|---|---|---|---|---|---|---|---|---|---|---|---|---|---|---|---|
| Method | why | which | who | what | when | where | how | others | why | which | who | what | when | where | how | others |
| FT | 0.48 | 0.38 | 0.43 | 0.42 | 0.40 | 0.38 | 0.47 | 0.41 | 0.46 | 0.36 | 0.37 | 0.42 | 0.38 | 0.38 | 0.39 | 0.39 |
| MEMIT | 0.89 | 0.75 | 0.76 | 0.75 | 0.79 | 0.93 | 0.79 | 0.76 | 0.49 | 0.42 | 0.46 | 0.43 | 0.45 | 0.46 | 0.48 | 0.48 |
| AlphaEdit | 0.94 | 0.83 | 0.88 | 0.86 | 0.93 | 0.83 | 0.88 | 0.88 | 0.49 | 0.49 | 0.49 | 0.50 | 0.50 | 0.50 | 0.49 | 0.49 |

Table 4: Performance of LLaMA-3.2 (3B) across different question types (reordered). Left: Editing Success Rate. Right: General Ability.

nism effectively insulates general capabilities from the specifics of the edit, regardless of the target's representational clarity.

### 5.3 IMPACT OF QUESTION TYPE.

We now turn to the role of **question type**, which reflects the linguistic structure of the edit prompt. Tables 3 and 4 present results across eight interrogative categories. Among them, **"Why"** and **"Which"** emerge as the most divergent, consistently defining the upper and lower bounds of editing success.

Across all models and editing algorithms, "Why" questions achieve the highest success rates. For example, with AlphaEdit on LLaMA-3.1, "Why" questions reach a success rate of 0.79, whereas "Which" questions fall to 0.70. A similar gap is observed for the smaller LLaMA-3.2 model, where AlphaEdit yields 0.94 on "Why" but only 0.83 on "Which". These discrepancies suggest that different interrogatives engage distinct representational and reasoning pathways inside the model.

We hypothesize that "Which" questions are particularly difficult because they require the model to select a single discrete option from a constrained set, thereby invoking rigid and competitive factual associations. Overwriting such tightly coupled representations likely causes stronger interference, which also explains why "Which" edits tend to produce larger drops in general ability for parameter-modifying methods like FT and MEMIT. In contrast, "Why" questions often elicit explanatory or causal reasoning, which draws on more distributed and semantically flexible representations. These representations appear more amenable to modification, yielding both higher success rates and reduced collateral damage.

Interestingly, AlphaEdit demonstrates relative robustness across question types: while performance still varies, its null-space projection mechanism ensures that side effects on general ability remain constant (0.55 for LLaMA-3.1 and 0.49 for LLaMA-3.2). This highlights the importance of algorithmic safeguards in mitigating the structural challenges posed by different question forms.

Taken together, these findings show that the **syntactic structure of the edit prompt is not a neutral choice**. Certain question types, particularly "Which", inherently pose greater risks for both efficacy and stability, underscoring the need for question-aware editing strategies.

## 6 VALIDATING THE KNOWLEDGE-DIAGNOSTIC FRAMEWORK

The preceding analysis establishes that knowledge characteristics are strong predictors of editing difficulty. Based on this, we proposed the Knowledge-Diagnostic Framework to improve performance by applying an intensive edit strategy only to cases diagnosed as "Hard." We define Hard Cases as knowledge that is "Known", "Unfamous", or of the "Which" type. We operationalize the intensive strategy as applying AlphaEdit 5 times. This section empirically validates this approach.

| Model | Popularity | | | | Familiarity | | | | Question Type | | | |
|---|---|---|---|---|---|---|---|---|---|---|---|---|
| | Hard (Unfamous) | | Easy (Famous) | | Hard (Known) | | Easy (Unknown) | | Hard (Which) | | Easy (Why) | |
| | 1x | 5x | 1x | 5x | 1x | 5x | 1x | 5x | 1x | 5x | 1x | 5x |
| LLaMA-3.1 | 0.82 | **0.87** | 0.88 | 0.88 | 0.84 | **0.86** | 0.88 | 0.88 | 0.70 | **0.75** | 0.79 | 0.80 |
| LLaMA-3.2 | 0.76 | **0.81** | 0.82 | 0.83 | 0.82 | **0.86** | 0.87 | 0.87 | 0.83 | **0.93** | 0.94 | 0.94 |

Table 5: Editing success rates across three dimensions of knowledge characteristics: **Popularity** (Famous vs. Unfamous), **Familiarity** (Known vs. Unknown), and **Question Type** (Which vs. Why). Results are reported for both LLaMA-3.1 (8B) and LLaMA-3.2 (3B), under single-edit (1x) and repeated-edit (5x) conditions.

| Metric | Baseline (5x for All) | Our Framework |
|---|---|---|
| Total Compute Time | 83.3 hours | **56.7 hours** |
| Total Cost | $50.00 | **$34.00** |
| **Efficiency Gain** | - | **32%** |

Table 6: Cost-benefit comparison of a naive intensive strategy versus our adaptive Knowledge-Diagnostic Framework.

## 6.1 EXPERIMENTAL VALIDATION

Tables 5 compares the success rates of a standard (1x) edit versus an intensive (5x) edit for our "Hard vs. Easy" pairs. The results consistently validate our hypothesis. In all three scenarios, the performance on Hard Cases benefits significantly from the intensive 5x edit, with success rates rising to match the performance of Easy Cases. For example, the success rate for the hard "Which" category on LLaMA-3.2 jumps from 0.83 to 0.93. Crucially, this elevated performance now matches the "Why" category, which was already at a performance plateau (0.94) and did not benefit from repeated edits. This demonstrates that the intensive strategy effectively closes the performance gap by overcoming the inherent difficulty of the hard cases, while avoiding wasted computation on easy cases.

## 6.2 COST-BENEFIT ANALYSIS

The value of our framework extends beyond efficacy to practical efficiency. We conducted a cost-benefit analysis based on editing the LLaMA 3.1 8B model on our 2000-item dataset, where 60% of items are diagnosed as "Hard." As summarized in Table 6, a naive approach of applying an intensive 5x edit to all items requires 83.3 hours of A6000 GPU time at a cost of $50.00. In contrast, our Knowledge-Diagnostic Framework, which applies intensive edits only to the 60% of hard cases, achieves a comparable level of performance in just 56.7 hours, costing $34.00. This represents a **32% reduction in both time and cost**, providing a clear economic incentive for adopting a knowledge-aware, adaptive editing strategy in large-scale applications.

## 7 CONCLUSION

We present a systematic study of the role that knowledge characteristics play in determining the outcomes of model editing. By introducing the *Knowledge Spectrum*, we show that dimensions such as popularity, familiarity, and question type are not peripheral factors but central predictors of editing success and stability. Our proposed *Knowledge-Diagnostic Framework* leverages these insights to adapt editing strategies according to the diagnosed difficulty of a knowledge item, thereby improving efficacy while reducing computational cost. Extensive experiments demonstrate that adaptive editing substantially narrows the performance gap between hard and easy cases, all while safeguarding the model's general reasoning capabilities. Beyond technical contributions, our findings highlight a broader lesson: editing is not merely an algorithmic problem but a knowledge-aware process. Future work should continue to expand evaluation protocols to capture long-form generation and general abilities, ensuring that model editing advances are both reliable and holistic. We hope this study provides a foundation for the development of safer, more efficient, and more interpretable editing protocols.

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
