# OpenReview forum: "Diagnosing Model Editing via Knowledge Spectrum"
_ICLR.cc/2026/Conference — ICLR 2026 Conference Withdrawn Submission_

### Official Review · Reviewer_84Kp · 2025-10-25

**Soundness:** 2
**Presentation:** 2
**Contribution:** 2
**Rating:** 2
**Confidence:** 4

**Summary:**

This paper studies how knowledge properties, including popularity, familiarity, and question type,  influence the difficulty of model editing. It proposes the Knowledge Spectrum to categorize editing targets and finds consistent differences in edit success across these dimensions. The authors further introduce a knowledge-diagnostic framework that adaptively adjusts editing intensity based on difficulty. Experiments on RealTimeQA show improved success on hard edits with lower compute cost.

**Strengths:**

- The paper proposed a Knowledge-Diagnostic Framework to explore knowledge editing performance empirically based on the properties of knowledge and questions.

**Weaknesses:**

- The novelty is limited. Only 4.2 and 4.3 describe the proposed methods, which are mostly conceptual, like a taxonomy or a simple conditional strategy, rather than methodological or algorithmic.
- The significant limitation of the proposed framework is that metadata of the knowledge to be edited is required, which is difficult to obtain. The diagnostic framework assumes access to knowledge-level labels, such as whether a fact is known vs. unknown to the model and famous vs. unfamous, which are not readily available in practical deployment. This reliance on externally determined or dataset-derived metadata limits applicability outside curated research settings.
- The study is positioned as model editing, yet the evaluation dataset, RealTimeQA, is a factuality QA benchmark rather than a standard editing benchmark. It is unclear whether the observed editing difficulty patterns would persist on canonical editing datasets (e.g., CountFact, ZsRE, MQuAKE).
- generality and reproducability: Only three editing baselines and one dataset are considered, which is lacking in generality. Without a reproducibility statement section, there are no detailed implementation details. For example, although Section 4.2 mentions using Wikipedia page views and SliCK-style probing to define “popularity” and “familiarity”, the paper does not specify critical implementation details, such as thresholds for binning, decoding strategies, evaluation criteria, or time windows for page-view collection. The diagnostic classification pipeline is described only conceptually, leaving the experiments difficult to reproduce and potentially dataset-specific. Meanwhile, no code or reproducibility statement is provided.

**Questions:**

- The framework assumes access to “known vs. unknown” and “famous vs. unfamous” labels. How would these metadata be obtained in a real deployment setting where they are not available?

- Since all experiments are conducted only on RealTimeQA, a factuality QA dataset rather than an editing benchmark, what evidence suggests that the findings would generalize to canonical editing datasets such as ZsRE or MQuAKE?

- Can the authors specify the exact thresholds and procedures used for popularity and familiarity classification (e.g., Wikipedia view cutoffs, probing method), so that the diagnostic step can be reproduced?

---

### Official Review · Reviewer_7mWY · 2025-10-26

**Soundness:** 1
**Presentation:** 1
**Contribution:** 1
**Rating:** 0
**Confidence:** 4

**Summary:**

The paper classifies target knowledge to be edited using three criteria: Popularity (via the number of Wikipedia page views), Familiarity (Known vs. Unknown, based on the SliCK criterion proposed in prior work), and Question Type (Who/What/When/Where/Which/Why/How/Others). It shows that edit success varies across these categories. Based on the edit success scores for each category under these criteria, the paper designates as “Hard” any knowledge that is Unfamous, Known, or of the Which type. For these hard cases, the authors apply the AlphaEdit method from prior work five times and demonstrate improved edit success, narrowing the gap with easier cases.

**Strengths:**

The paper empirically analyzes how three criteria—popularity, familiarity, and question type—affect editing performance. Notably, its quantitative analysis by question type fills a gap in prior work. Building on these findings, the authors propose a simple strategy for harder cases: reapplying prior editing methods up to five times. This yields consistent gains primarily in the Hard cases, with little to no benefit in Easy cases, suggesting that the approach meaningfully targets the difficult regime rather than providing uniform improvements.

**Weaknesses:**

# 1. Novelty Inflation
## a) Contribution 1 (as written in Introduction, line 060-61)
Among the three “knowledge spectrum” dimensions listed, popularity and familiarity are not novel to this work. Prior studies have already analyzed popularity/exposure in depth [1, 2] and even found that famous entities are often harder to edit, which puts the paper’s directional claim at odds with established findings. Likewise, for familiarity (known vs. unknown), prior work has examined whether a model already knows a fact—often framed as injecting an unknown fact [1].
## b) Contribution 2 (as written in Introduction, line 062-063)
Although the paper claims to “broaden evaluation beyond locality”, it in fact omits standard locality assessment. Instead, it treats assessing post-edit performance on general-ability benchmarks as that broadening. However, prior works evaluate locality and, in addition, assess performance on a wide range of general-ability benchmarks. Notably, AlphaEdit—which this paper itself uses—is a representative prior method that conducts deep evaluations across diverse general-ability benchmarks.
## c) Contribution 3 (as written in Introduction, line 064-066)
The paper adopts a setting where each edit is applied five times and then claims it “saves compute” on Easy cases by applying five times only to Hard cases. But established methods are designed and evaluated as single-iteration editing. Since the paper first inflates the baseline cost (5x edits everywhere) and then presents a “saving” relative to that inflated baseline, calling the resulting difference a saving is overstated relative to standard practice.
## References
 [1] Evaluating the Ripple Effects of Knowledge Editing in Language Models (TACL 2024). https://aclanthology.org/2024.tacl-1.16.pdf

 [2] Is it Possible to Edit Large Language Models Robustly? (ICLR 2024 Workshop LLMAgents). https://openreview.net/forum?id=dZ2VW5gp5g


# 2. Critical Omissions in Methods and Results
Too many essential specifications are missing to enable understanding, replication, or verification.

## a) Popularity (Famous vs. Unfamous) Definition Missing
The paper claims to use page-view–based popularity but provides no numeric criteria, statistical analysis or thresholds for splitting “Famous” vs. “Unfamous”.

## b) Familiarity (Known vs. Unknown) Procedure Unspecified
Beyond noting it is “inspired by SliCK”, the paper gives no concrete protocol. The SliCK paper states a few-shot setup, yet this work does not describe few-shot configuration, decoding strategies actually used, or decision thresholds for labeling Known vs. Unknown.

## c) Dataset Choice and Construction (RealTimeQA) Missing Rationale and Pipeline
- Despite well-established editing benchmarks (e.g., CounterFact, zsRE), the paper preprocesses RealTimeQA, which is not a knowledge-editing dataset, without explaining why it is preferable.
- The preprocessing pipeline is unreported: how entities were extracted, how the target fact to “edit” was determined for each prompt.
- It claims hand-written paraphrases for generalization but provides neither a QC protocol nor even minimal examples.
- Prompts (and evaluation) for locality are neither constructed nor reported, with no justification for this omission.

## d) Editing Setup Core Details Missing
Missing: the total number of edits, whether edits were applied sequentially or in batch, and any ordering/constraint rules.

## e) Five Passes on Hard Cases Justification and Ablations Missing
The choice to apply AlphaEdit exactly five times to “Hard” cases is unmotivated. There is no ablation over the number of passes (e.g., 1–6+) to identify an optimal setting or to quantify efficacy vs. side effects.

## f) General Ability for Five Times Editing Results Omitted
In Table 5, the 5-times setting reports only edit success. General Ability is omitted, which is especially concerning given this non-standard multi-apply setup.

**Questions:**

1. My questions follow directly from the omissions identified in Weaknesses (2. Critical Omissions in Methods and Results). Those missing elements are precisely what I need clarified for understanding and verification.
2. As a suggestion, rather than always applying five passes to Hard cases, a stronger contribution would be an adaptive framework that estimates difficulty and selects the optimal number of applications per instance. Include a principled stopping rule and ablations that demonstrate the trade-off between edit success and side effects, so that the procedure chooses an optimal pass count instead of fixing it.

---

### Official Review · Reviewer_e3C4 · 2025-10-29

**Soundness:** 2
**Presentation:** 2
**Contribution:** 2
**Rating:** 2
**Confidence:** 4

**Summary:**

This paper presents the "Knowledge Spectrum," a three-dimensional framework for categorizing knowledge and predicting the difficulty of editing factual information within pretrained language models. The dimensions—popularity, familiarity, and question type—are shown to impact both editing success and safety. The authors then propose the "Knowledge-Diagnostic Framework," an adaptive method to tailor the intensity of model editing efforts based on a diagnosed difficulty profile, aiming for greater efficiency and reliability. Through systematic empirical studies, the paper demonstrates the predictive power of these dimensions and the effectiveness of adaptive editing on real-world benchmarks.

**Strengths:**

* S1: The motivation is reasonable: the authors attempt to explore when and why certain model edits succeed or fail, which is an important but underexplored question in the field of model editing.

* S2: The proposed adaptive framework is simple and computationally efficient — tailoring editing effort to “hard” vs. “easy” cases could be practically useful if validated properly.

**Weaknesses:**

* W.1: Limited novelty: The proposed "Knowledge-Diagnostic Editing Framework" is mainly a heuristic combination of existing ideas — diagnostic probing (from SliCK), locate-and-edit editing (from MEMIT/ROME/AlphaEdit), and simple conditional repetition. There is no new algorithm, theory, or optimization principle. The contribution is incremental and could be viewed as a survey-style synthesis.

* W.2: Conceptual framework lacks depth: The "Knowledge Spectrum" is an intuitive taxonomy, but it does not introduce new insights beyond prior notions of popularity and familiarity that already exist in knowledge probing literature.

* W.3: Weak methodological justification: The adaptive framework simply applies more editing iterations for "hard" cases. This strategy is neither learned nor optimized — it is a manually set rule. Consequently, the paper’s claimed "framework" reduces to an empirical tuning policy.

* W.4: Experimental validation is shallow: Experiments reuse AlphaEdit almost directly and evaluate only on a few benchmarks without rigorous statistical testing or ablations. And the relation between famous and editing is also discussed in previous work [1]. Also, General Ability is also evaluated only on relatively limited datasets.


$$Ref:$$

1. Is it Possible to Edit Large Language Models Robustly?, 2024.

**Questions:**

* Q1: I'm particularly curious about how the three dimensions of the "Knowledge Spectrum" are selected. For instance, the "Question Type" dimension appears to consider only interrogative sentences, yet a significant portion of knowledge editing data begins with declarative sentences: `The name of the country which Goursez Vreizh is associated with is` in [1].

* Q2: The author's assessment of knowledge editing damage fails to convince me. Could the author supplement with additional experiments, such as changes in model norm after editing [2-4]?



$$Ref:$$
[1] KnowEdit: A Benchmark of Knowledge Editing for LLMs, 2024.

[2] Model Editing at Scale leads to Gradual and Catastrophic Forgetting, 2024.

[3] Reasons and Solutions for the Decline in Model Performance after Editing, 2024.

[4] Model Editing Harms General Abilities of Large Language Models: Regularization to the Rescue, 2024.

---

### Official Review · Reviewer_2tP6 · 2025-10-31

**Soundness:** 3
**Presentation:** 3
**Contribution:** 3
**Rating:** 6
**Confidence:** 4

**Summary:**

The paper introduces the Knowledge Spectrum, a framework for understanding how intrinsic properties of target knowledge—popularity, familiarity, and question type—affect model editing outcomes. It then proposes a Knowledge-Diagnostic Framework, an adaptive strategy that adjusts the intensity of model editing (using AlphaEdit) based on these knowledge characteristics.

**Strengths:**

**1. Novel perspective**

The paper shifts the focus of model editing research from algorithmic optimization to knowledge-level diagnostics, offering a fresh and interpretable framework.

**2. Systematic analysis**

The Knowledge Spectrum provides a structured way to dissect how intrinsic knowledge properties affect editability and stability—an underexplored but essential dimension.

**3. Practical contribution**

The Knowledge-Diagnostic Framework is simple yet effective, improving editing success while reducing computation by ~32%. This makes the approach practical for large-scale applications.

**Weaknesses:**

## Analysis Framework

**1. Dimension overlap in the Knowledge Spectrum framework**

The dimensions Familiarity and Popularity appear conceptually overlapping. Intuitively, LLMs tend to be more familiar with popular entities. The authors should conduct a comparative analysis and provide a clearer explanation of how these two dimensions differ both conceptually and empirically.

**2. Threshold for defining “Famous” vs. “Unfamous”**

The paper uses Wikipedia page views to categorize popularity, but it is unclear what numerical thresholds define “high” or “low.” Are these thresholds domain-dependent? The authors should clarify the exact criteria and discuss whether the classification is stable across domains.

**3. Criteria for defining “Known” vs. “Unknown”**

The determination of whether knowledge is “known” to the model is underspecified. Since the model’s responses can vary with different prompts or decoding strategies, what specific probing setup or decision rule defines an item as Known? The authors should elaborate on how they ensure consistency across prompts and decoding conditions.

## Experiments

**1. Limited to parameter-modifying methods**

The experiments only consider two methods (MEMIT and AlphaEdit), both of which are parameter-modifying. It remains unclear whether the proposed analysis and framework apply to parameter-preserving approaches (e.g., adapter-based or memory-augmented methods) discussed in the related work.

**2. Model diversity**

The experiments are exclusively conducted on LLaMA models. It would strengthen the paper to test whether the proposed Knowledge-Diagnostic strategy generalizes to other architectures such as Qwen or Mistral, which may differ in internal representation and editing sensitivity.

**3. Dataset limitation**

RealTimeQA is the only dataset used for evaluation. The applicability of the proposed strategy to other knowledge-editing benchmarks (e.g., CounterFact, ZsRE, or factual datasets in other domains) remains unclear.

**4. Independent treatment of knowledge dimensions**

The paper analyzes the impact of Familiarity, Popularity, and Question Type separately. However, in real scenarios, these factors often interact. It would be valuable to include experiments on combined conditions (e.g., editing known–unfamous–which-type items) to understand joint effects.

## Writing and Organization

The evaluation metrics are repeatedly described in Sections 3.2, 4.1, and 4.4. This repetition affects readability. I recommend consolidating the metric definitions in Section 3.2 and referring back to them later instead of reintroducing them.

**Questions:**

seed weaknesses

---

### Note · Authors · 2025-12-05

I have read and agree with the venue's withdrawal policy on behalf of myself and my co-authors.